# LEARNING BEYOND PROXIMITY: CAUSAL REASONING WITH LLMS FOR ROBUST POI PREDICTION

## ABSTRACT

Point-of-Interest (POI) prediction forecasts a user's next destination from mobility history. A key challenge is geographic exposure bias, where users often visit nearby or popular places out of convenience rather than genuine interest. Such convenience-driven behaviors create spurious correlations that obscure true preferences, leading models to misinterpret frequent check-ins as strong signals of interest. Traditional sequential/graph models rely on surface-level statistical correlations, and recent Large Language Model (LLM)-based methods improve semantic coverage but still inherits exposure bias from observational logs. We address this with causal inference, explicitly modeling the data-generating process and distinguishes preference-driven behaviors from convenience-driven ones. In particular, we estimate geographic propensity scores that quantify the likelihood of a visit due to spatial exposure, and use them to reweight check-ins and align trajectory retrieval in exposure-consistent space. Towards this end, we propose Causal Geographic Prediction (CGP), a unified framework that integrates causal inference with LLM-based trajectory modeling. It employs exposure-aware trajectory prompting, causal-geographic similarity alignment, and supervised fine-tuning to separate genuine preferences from convenience-driven behaviors. Experiments on real-world datasets show that CGP outperforms state-of-the-art baselines.

## 1 INTRODUCTION

Point-of-Interest (POI) prediction aims to forecast a user's next destination from historical check-in patterns (Acharya & Mohbey, 2024; Islam et al., 2022), enabling applications such as personalized navigation and urban mobility analysis (Qi et al., 2018). A central challenge is to separate true user preferences from visits driven primarily by geographic convenience(Psyllidis et al., 2022; Ying et al., 2012). In practice, users often check in at places close to home, work, or daily routes—not because they genuinely like them, but because they are easy to access. For example, a nearby café on the way work may be visited far more often than a distant café, even though the latter better reflects the user's actual interest (Figure1). Such patterns create geographic exposure bias, where spatial proximity influences both historical trajectories and future predictions. As a result, existing models tend to conflate convenience-driven visits with genuine preferences, leading to biased and less interpretable recommendations. This motivates our research question: **How can we distinguish genuine user interests from proximity-induced behaviors for more robust POI predictions?**

Early research on POI prediction explored diverse paradigms, including matrix factorization (Rendle et al., 2010), recurrent neural networks such as LSTMs (Hochreiter & Schmidhuber, 1997), and attention-based architectures (Luo et al., 2021). These models captured spatial and temporal dependencies in check-in sequences but still relied on surface-level correlations such as visit frequency. As a result, they often mistook convenience-driven visits for true preferences while overlooking less frequent but more meaningful ones. Recent advances leverage Large Language Models (LLMs) to transform structured check-in data into natural language prompts (Li et al., 2024a; Liu et al., 2025; Wang et al., 2024), fusing semantic, temporal, and spatial signals into a unified representation. Although more expressive, LLM-based methods still operate on observational data and remain vulnerable to geographic exposure bias (Morehouse et al., 2024). Thus, merely scaling model capacity cannot address the core challenge of disentangling genuine preferences from spurious correlations induced by spatial convenience.

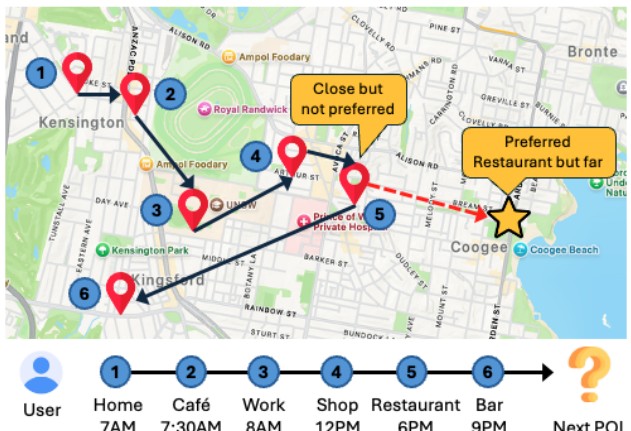

Figure 1: An example of geographic exposure bias: user may often visit a nearby place out of convenience along daily route, while a less visited but more distant place may better reflect true preference.

Towards this end, we introduce causal inference as a principled way to reduce geographic exposure bias (Yu et al., 2023; Pearl, 2009). In observational mobility data, user visits are shaped not only by genuine preferences but also by confounders—factors that jointly affect exposure and observed behavior. Factors such as distance affect both a user's exposure to a POI and their likelihood of visiting it, misleading even advanced models to interpret convenience as genuine preference. These confounders create spurious correlations, making frequent nearby check-ins appear as strong preferences, even when advanced LLMs are applied. Causal inference explicitly models the data-generating process, allowing us to adjust for these confounders so that visits at different exposure levels can be compared fairly, as if randomized (Imbens & Rubin, 2015; Burgess & Thompson, 2021). As a result, observed behavioral differences are more likely to reflect true preferences rather than artifacts of spatial convenience. To operationalize this, we estimate a geographic propensity score (Thoemmes & Kim, 2011) for each check-in, quantifying the probability of visiting a POI under given spatial and contextual conditions. These scores act as balancing variables that down-weight convenience-driven visits and amplify more indicative preference signals. They also refine trajectory similarity by penalizing cases where trajectories are semantically alike but differ substantially in exposure conditions. Through this causal adjustment, we construct LLM prompts from exposure-consistent histories, enabling the model to better capture genuine user interests.

In this work, we propose Causal Geographic POI Prediction (CGP), a unified framework that integrates causal inference with LLMs to achieve robust POI prediction. CGP mitigates spatial confounding by embedding causal adjustments into both input construction and model adaptation, allowing the LLM to distinguish true user interests from proximity-induced behavior. Specifically, CGP includes five components: (1) Trajectory Prompting transforms each user's check-in sequence into structured natural language prompts, encoding rich spatio-temporal semantics; (2) Geographic Propensity Estimation employs a neural embedding-based model to estimate the propensity score for each check-in, quantifying the likelihood of visiting a POI due to spatial exposure; (3) Causal-Geographic Similarity Computation defines semantic similarity by penalizing differences in propensity scores, ensuring retrieved examples are consistent in both behavioral intent and exposure context; (4) Historical Trajectory Selection retrieves auxiliary trajectories using the adjusted similarity, providing exposure-consistent examples for model input; (5) Supervised Fine-Tuning adapts the pretrained LLM to these prompts via parameter-efficient updates, enabling preference modeling disentangled from proximity bias. Experimental results on real-world datasets show that CGP outperforms state-of-the-art POI models. Our main contributions are summarized as follows:

- To the best of our knowledge, this is the first work to embed causal inference into LLM-based POI prediction, explicitly mitigating exposure bias from a geographic perspective for more robust performance.

- We design an embedding-based *Geographic propensity score* that estimates the likelihood of visiting a POI from its spatial and contextual attributes, serving as a causal balancing variable to separate genuine preferences from convenience-driven behaviors.

- We develop a retrieval–prompting strategy that aligns historical trajectories by geographic exposure, allowing the LLM to separate preferences from convenience-driven behaviors.

## 2 RELATED WORK

POI prediction is a core task in location-based social networks (LBSNs), aiming to forecast a user's next destination from historical mobility and contextual signals (Yin et al., 2016; Islam et al., 2022). Early methods, such as collaborative filtering and matrix factorization, modeled user–POI interactions but largely ignored explicit spatial–temporal dependencies (Davtalab & Alesheikh, 2021; Yang et al., 2017). Subsequent research adopted sequential models, including Markov chains and recurrent neural networks, to capture trajectory dependencies (Altaf et al., 2018; Mathivanan et al., 2024), while more recent work employed attention and graph neural networks to model higher-order relations among users, POIs, and contexts (Ni et al., 2024; Yu et al., 2024a). Despite these advances, a key challenge remains: separating genuine user preferences from behaviors driven by geographic convenience or popularity. Traditional and even recent models often overweight nearby or frequently visited POIs, leading to exposure bias that distorts preference estimation. Geographic and temporal cues have been incorporated to alleviate this bias, yet these features alone cannot fully account for the confounding effect of proximity, leaving residual bias in predictions. The emergence of LLMs offers a new direction, enabling structured check-ins to be transformed into natural language prompts that integrate semantic, temporal, and spatial features into unified representations (Kumar, 2024; Ji et al., 2025). Enhancements with explicit spatial or temporal cues have improved mobility understanding (Ding & Wang, 2025; Yuan et al., 2025), but LLM-based POI recommenders remain susceptible to proximity-driven confounding (Hwang et al., 2018).

Causal inference offers a principled framework for disentangling genuine effects from spurious correlations caused by confounders. Foundational methods such as the back-door criterion and propensity score adjustment (Pearl, 2009; Yu et al., 2025) are widely used to estimate treatment effects from observational data, particularly in fields like healthcare and economics. In recommender systems, causal inference has become an effective tool to address biases including popularity, exposure, and temporal confounding (Yao et al., 2021). By modeling the data-generating process as a causal graph, these approaches adjust for variables that jointly affect both user preferences and observed actions, resulting in more accurate and interpretable predictions. Recently, causal methods have been adapted to POI prediction tasks through techniques such as propensity score matching, causal regularization, and counterfactual reasoning, aimed at correcting spatial and contextual biases in mobility data (Yu et al., 2023; Zeng et al., 2022). However, the integration of causal inference into large-scale language models is still in its infancy. This work addresses that gap by embedding causal adjustments—specifically, geographic propensity scores—within LLM-based POI prediction, enabling robust preference estimation.

## 3 PROBLEM DEFINITION

POI prediction aims to forecast a user's next destination based on historical check-ins within a location-based social network (LBSN) (Yin et al., 2016; Acharya & Mohbey, 2024). Let $\mathcal{D} = \{q_i\}_{i=1}^N$ be a dataset of $N$ check-ins, where each $q_i = (u_i, p_i, c_i, t_i, g_i)$ denotes a user ID $u_i$, POI identifier $p_i$, category $c_i$, timestamp $t_i$, and geographic coordinates $g_i \in \mathbb{R}^2$. For each user $u$, check-ins are ordered and segmented into trajectories $T_u = \{T_u^{(m)}\}_{m=1}^{L_u}$ using a temporal window $\Delta t$. Given a partial trajectory $T_u^{(m)}(t) = \{(p_\ell, c_\ell, t_\ell, g_\ell)\}_{\ell=1}^k$ with $t_1 < \cdots < t_k \leq t$, the task is to predict the next POI $pk + 1$ at time $t_{k+1}$. A key challenge is geographic exposure bias (Psyllidis et al., 2022; Ying et al., 2012), where spatial proximity shapes both mobility history and future visits. Users often visit nearby POIs out of convenience, causing models to misinterpret convenience-driven behavior as true preference. Yet, true interests may lie in distant POIs that remain unobserved in raw records due to spatial constraints, creating a gap between observed check-ins and actual preferences.

To address this, we model the data-generating process from a causal perspective. As shown in Figure 2, a user's true preference $P$ influences the likelihood of visiting a POI $V$, while geographic distance $D$ acts as a confounder, affecting both historical trajectories $H$ and visit outcomes. Temporal variables $X$ may also shape both preferences and visit behavior. This structure exposes two back-door paths: (1) $P \leftarrow H \leftarrow D \rightarrow V$, where distance influences both mobility history and visit

behavior, making frequent visits to a nearby mall reflect convenience rather than true interest; (2) $P \leftarrow X \rightarrow V$, where temporal factors (e.g., morning check-ins) bias preferences toward specific POI types. These confounding paths create spurious correlations that bias preference estimation. We therefore estimate a geographic propensity score for each check-in, quantifying the probability of visiting a POI given spatial and contextual attributes. This score serves as a causal balancing factor, mitigating exposure bias and enabling more robust POI prediction.

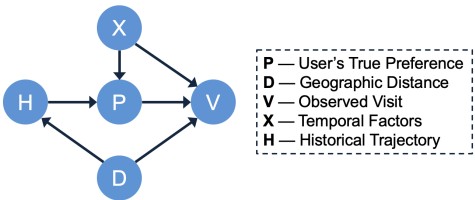

Figure 2: Our designed causal graph representing the data-generating process in POI predictions.

## 4 METHODOLOGY

As shown in Figure 3, our CGP framework builds on LLMs as the backbone, as they effectively unify heterogeneous signals into a single representation. Yet applying LLMs directly to location data introduces exposure bias, mistaking nearby visits for genuine preferences. To address this, CGP embeds causal adjustments into data preparation and training, enabling the framework to separate convenience-driven from true preference-driven choices. CGP consists of five components: (1) **Trajectory Prompting** converts raw check-in sequences into structured natural language prompts, enabling the LLM to jointly capture spatial, temporal, and semantic signals; (2) **Geographic Propensity Estimation** models the likelihood (e.g., propensity score) of visiting a POI based on spatial exposure using an embedding-based network; (3) **Causal-Geographic Similarity Computation** adjusts semantic similarity by penalizing differences in propensity scores, ensuring retrieved trajectories share both behavioral patterns and exposure conditions; (4) **Historical Trajectory Selection** retrieves auxiliary trajectories using the adjusted similarity, forming exposure-consistent prompts for model input; (5) **Supervised Fine-Tuning** adapts a pretrained LLM to these prompts through parameter-efficient updates, enabling preference modeling disentangled from spatial confounding.

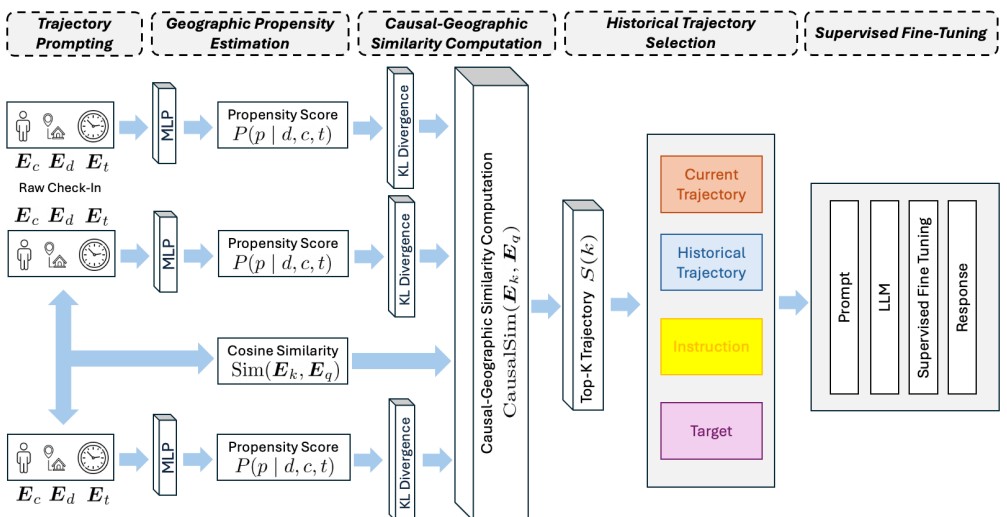

Figure 3: Overall framework of our proposed method.

### 4.1 TRAJECTORY PROMPTING

The first challenge in POI prediction is representing heterogeneous spatio-temporal data (Zhao et al., 2020). As shown in Figure 2, trajectories are shaped by both true preferences and confounders

such as geographic distance. To model preferences accurately, the representation should capture behavioral intent while accounting for these confounders, as ignoring them can bias predictions. Traditional numerical embeddings often miss such nuances, limiting performance in geographically diverse settings (Yu et al., 2024b; Hu et al., 2024). We address this by converting structured check-in records into natural language, enabling LLMs to jointly interpret POI categories, timestamps, and locations through semantic reasoning. This unified, context-aware format lays the groundwork for incorporating causal signals, improving robustness to spatial confounding.

Formally, for a user's historical trajectory $T_u^j = (p_1, c_1, t_1, g_1), \ldots, (p_k, c_k, t_k, g_k)$, each check-in is converted into a natural language sentence using a predefined template that encodes semantic, temporal, and spatial attributes. The overall prompt and check-in record structure is shown in Table 1. Each prompt includes four blocks: the current trajectory block encodes the user's recent check-ins to capture short-term behavioral context; the historical trajectory block contains trajectories retrieved via causal-geographic similarity to ensure consistency in both semantic behavior and geographic exposure; the instruction block specifies the prediction task, guiding the model to prioritize preference-driven patterns over proximity effects; and the target block, used only during training, provides the ground-truth POI to help the model learn predictions disentangled from spatial confounding.

Table 1: Prompt template for trajectory construction and check-in representation.

| Prompt Layout |
| --- |
| [Current Trajectory]: The user [user id] has recently checked in at: [records]. |
| [Retrieved Trajectories]: Historical sequences from similar comparable contexts: [records]. |
| [Task Instruction]: Using both the recent and retrieved visits, infer the POI the user is most likely to favor next, emphasizing intrinsic preference over geographical convenience. |
| [Target Answer]: At [time], user [user id] visited at [poi id]. |
| **Check-in Sentence Pattern** |
| At [time], user [user id] visited [poi id] (type: [category]), located [distance] meters from the prior check-in, with an estimated propensity score of [value]. |

Our trajectory prompting template transforms heterogeneous mobility logs—covering spatial, temporal, and semantic attributes—into a unified textual form suitable for LLMs. By clearly distinguishing the recent context, exposure-consistent historical sequences, the task directive, and the target outcome, the template facilitates context-aware and bias-resilient modeling. In addition, the check-in representation encodes both the raw confounder (distance) and its adjustment (geographic propensity score). For instance, each record $q = (u, p, c, t, g)$ is expressed as: "At [time], user [id] visited [poi] (category: [c]) located [distance] meters from the last check-in, with a propensity score of [value]." This design lets the model interpret both exposure intensity and its causal correction, thereby distinguishing habitual proximity-driven actions from genuine user interests. By embedding multimodal cues into natural language, our approach preserves causal relationships and enhances interpretability under spatial confounding.

## 4.2 GEOGRAPHIC PROPENSITY ESTIMATION

Although spatial distance is included in trajectory prompts, raw distance alone cannot capture its systematic influence on mobility. Geographic proximity acts as a confounder, shaping both the likelihood of visiting a POI and the trajectory history, thereby obscuring true preferences with convenience-driven behaviors (Wang et al., 2025). To mitigate this, we estimate a geographic propensity score for each check-in, quantifying the probability of visiting a POI given its distance, category, and temporal context. This score serves as a causal balancing variable, aligning trajectories across semantic and exposure space to disentangle preference-driven patterns from spatial confounding.

For each check-in $q = (u, p, c, t, g)$, we estimate a geographic propensity score. The score models the probability of the treatment (i.e., a visit to POI $p$) conditioned on a set of covariates, including the spatial distance from the previous location, the POI category, and the temporal context. While logistic regression is commonly used for its simplicity and interpretability (Westreich et al., 2010),

it assumes linear and monotonic effects that seldom hold in real-world mobility, where temporal and categorical influences are often highly nonlinear. To capture such complexities, we employ an embedding-based neural architecture (Balaneshin-Kordan & Kotov, 2018) in which distance (discretized into bins), category, and temporal features are each mapped to trainable low-dimensional embeddings $\mathrm{Emb}(d)$, $\mathrm{Emb}(c)$, and $\mathrm{Emb}(t)$. These embeddings are concatenated and fed into a multi-layer perceptron, with a sigmoid activation producing the final propensity score as

$$P(p \mid d, c, t) = \sigma\left(f\left(\mathrm{Emb}(d), \mathrm{Emb}(c), \mathrm{Emb}(t)\right)\right) \tag{1}$$

where $f(\cdot)$ is the MLP and $\sigma(\cdot)$ is the sigmoid function. This architecture enables flexible, nonlinear mappings from contextual conditions to visit probabilities. The propensity score $P(p \mid d, c, t)$ quantifies the likelihood that a check-in is driven by spatial exposure rather than intrinsic preference.

### 4.3 Causal-Geographic Similarity Computation

Traditional semantic similarity measures capture behavioral patterns but often ignore spatial exposure differences, which can lead to misleading trajectory matches (Yan et al., 2013). Accordingly, we propose a causal-geographic similarity metric that jointly considers semantic relevance and exposure consistency. Generally, we refine cosine similarity between trajectory embeddings by penalizing divergence in their estimated geographic propensity score distributions, thereby discouraging matches between semantically similar but contextually mismatched trajectories.

Specifically, each trajectory prompt is first transformed into a natural language sentence and encoded into a dense embeddings using a pretrained LLM. Let $\boldsymbol{E}_k$ and $\boldsymbol{E}_q$ denote the embeddings of the key (current) and query (historical) trajectories, obtained from the LLM's final hidden layer. Mathematically, the baseline semantic similarity between two trajectory prompts is computed as

$$\mathrm{Sim}(\boldsymbol{E}_k, \boldsymbol{E}_q) = \frac{\boldsymbol{E}_k \cdot \boldsymbol{E}_q}{\|\boldsymbol{E}_k\|\|\boldsymbol{E}_q\|} \tag{2}$$

where $\mathrm{Sim}(\boldsymbol{E}_k, \boldsymbol{E}_q)$ is the cosine similarity between the embeddings of the key and query trajectories, capturing behavioral alignment but ignoring geographic exposure mismatches. To address this, we define a causal-geographic similarity that penalizes exposure discrepancies. Instead of a single scalar, each trajectory is represented as a distribution over its check-in–level geographic propensity scores, with divergence measured via KL divergence $D(P_k, |, P_q)$ to obtain the final similarity:

$$\mathrm{CausalSim}(\boldsymbol{E}_k, \boldsymbol{E}_q) = \mathrm{Sim}(\boldsymbol{E}_k, \boldsymbol{E}_q) \cdot \frac{1}{1 + \lambda \cdot D(P_k \,\|\, P_q)} \tag{3}$$

Here, $\mathrm{CausalSim}(\boldsymbol{E}_k, \boldsymbol{E}_q)$ denotes the causal-geographic similarity score, which integrates both semantic closeness and exposure consistency between the two trajectories. To compute exposure consistency, we construct empirical exposure distributions $P_k$ and $P_q$ for the key and query trajectories by collecting vector-valued propensity scores across their respective check-in sets $T_k$ and $T_q$:

$$P_k = \{P(p_q \mid d_q, c_q, t_q) \mid q \in T_k\}, \quad P_q = \{P(p_q \mid d_q, c_q, t_q) \mid q \in T_q\} \tag{4}$$

where the discrepancy function $D(\cdot \,\|\, \cdot)$ quantifies the divergence between these distributions, reducing similarity for trajectories that are semantically close but contextually mismatched. The hyperparameter $\lambda > 0$ controls the strength of this penalty, allowing the model to balance semantic similarity against exposure consistency.

### 4.4 Historical Trajectory Selection

Constructing effective prompts requires retrieving historical trajectories consistent in both semantic behavior and geographic exposure (Li et al., 2024b). If retrieval relies only on semantic similarity, trajectories may share surface patterns but differ in exposure, reintroducing proximity-related confounding. Accordingly, we use causal-geographic similarity to retrieve trajectories consistent in user intent and spatial context, ensuring prompts contain causally valid examples that guide the model toward preference-driven rather than proximity-induced predictions.

Given a historical trajectory pool $\mathcal{H}$, where each candidate trajectory $T_q$ ends before the current timestamp $t_k$, we compute its causal-geographic similarity with the current trajectory $T_k$. The top-$k$ most similar trajectories are selected as:

$$S(k) = \arg\max_{T_q \in \mathcal{H}, t_q^{\mathrm{end}} < t_k}^{k} \mathrm{CausalSim}(\boldsymbol{E}_k, \boldsymbol{E}_q) \tag{5}$$

where $t_q^{\text{end}}$ is the end time of $T_q$. $S(k)$ denotes the set of top-$k$ trajectories most aligned with $T_k$ in both intent and exposure. Selected trajectories are templated into natural language and added to the history block, ensuring structural consistency and exposure-aware conditioning so the model learns preference-driven signals instead of proximity-induced noise.

### 4.5 SUPERVISED FINE-TUNING

While the base model captures general language patterns, it lacks domain-specific knowledge of our causal adjustments. Fine-tuning with exposure-aware prompts aligns model parameters with the task's causal structure, enabling the model to distinguish preference-driven behavior from proximity bias. To reduce training cost while preserving performance, we adopt LoRA (Hu et al., 2022), freezing dense layers in the LLM and updating weights via low-rank decomposition. For a pretrained weight matrix $W_0 \in \mathbb{R}^{d \times k}$, we replace the full update $W_0 + \Delta W$ with:

$$W = W_0 + AB, \quad A \in \mathbb{R}^{d \times r}, \ B \in \mathbb{R}^{r \times k}, \ r \ll \min(d, k) \tag{6}$$

where $A$ projects from the original dimension $d$ to a reduced dimension $r$, and $B$ projects back to $k$. During training, only $A$ and $B$ receive gradient updates, reducing trainable parameters while retaining adaptability. For long prompts with multiple historical trajectories, we use FlashAttention-3 (Shah et al., 2024) for memory-efficient attention over sequences up to 32768 tokens.

## 5 EXPERIMENTS

To evaluate CGP, we conduct extensive experiments to answer the following questions: 1) Does our proposed CGP outperform existing state-of-the-art POI models? 2) What is the contribution of each component to the overall effectiveness of CGP? 3) To what extent does CGP enhance interpretability in POI task? 4) How sensitive is CGP to variations in its key hyperparameters?

### 5.1 SETUP

We conduct experiments on three widely used datasets: *Gowalla*[1] , *NYC-T*[2] , and *Ma-ST*[3] . Each contains user check-ins with POI categories, timestamps, and geographic coordinates. Following common practice, we filter out users with fewer than 10 check-ins and POIs visited by fewer than 10 users (Wu et al., 2025; Li et al., 2023). The remaining data are segmented into trajectories with temporal window $\Delta_t$, where the last POI is the prediction target and preceding POIs form the input. Our model is implemented in PyTorch and optimized with Adam (Modoranu et al., 2024) at learning rate 0.001. The embedding dimension is 128, batch size 256, and hyperparameters tuned via grid search. Each experiment is repeated five times with different seeds, and averages are reported. We use three LLM backbones (Touvron et al., 2023; Feng et al., 2024): Llama 3.3 (70B), Llama 4 Scout (17B), and Llama 2–7B. Performance is evaluated using Accuracy@1 (Acc@1), the proportion of cases where the ground-truth POI is ranked first: $\text{Acc@1} = \frac{1}{m} \sum_{i=1}^{m} \mathbb{I}(\text{rank}_i \leq 1)$, where $m$ is the number of test instances and $\text{rank}_i$ the predicted rank of the ground-truth POI.

We compare CGP against 10 representative baselines. **FPMC** (Rendle et al., 2010) combines matrix factorization with first-order Markov chains for preference and transition modeling, while **PRME** (Feng et al., 2015) learns pairwise ranking metric embeddings for user-specific movement patterns. **LSTM** (Hochreiter & Schmidhuber, 1997) captures long- and short-term dependencies, with **HST-LSTM** (Kong & Wu, 2018) extending it to a hierarchical structure for multi-scale spatial–temporal patterns. **STAN** (Luo et al., 2021) employs bi-layer attention for fine-grained spatio-temporal relevance, and **PLSPL** (Wu et al., 2020) uses parallel LSTMs to jointly capture long- and short-term preferences. **GETNext** (Yang et al., 2014) integrates a graph-enhanced transformer with trajectory flow mapping for global dependencies, while **STHGCN** (Yan et al., 2023) applies hypergraph convolution for high-order mobility and sparsity mitigation. **LLM4POI** (Li et al., 2024a) leverages structured trajectory prompts for heterogeneous check-ins, and **GA-LLM** (Liu et al., 2025) injects geographic coordinates and POI alignment for improved spatial representation.

---

[1] https://snap.stanford.edu/data/loc-gowalla.html

[2] https://sites.google.com/site/yangdingqi/home/foursquare-dataset?utm_source=chatgpt.com

[3] https://github.com/cruiseresearchgroup/Massive-STEPS

Table 2: (a) Performance comparison of Acc@1 (RQ1) and (b) case study (RQ3).

(a) Performance comparison (RQ1)

| Model | Gowalla | NYC-T | Ma-ST |
|---|---|---|---|
| FPMC | 0.101 | 0.083 | 0.128 |
| LSTM | 0.134 | 0.141 | 0.182 |
| PRME | 0.119 | 0.112 | 0.165 |
| HST-LSTM | 0.148 | 0.153 | 0.194 |
| PLSPL | 0.174 | 0.168 | 0.211 |
| STAN | 0.198 | 0.185 | 0.233 |
| GETNext | 0.217 | 0.206 | 0.248 |
| STHGCN | 0.246 | 0.233 | 0.275 |
| LLM4POI | 0.292 | 0.303 | 0.322 |
| GA-LLM | 0.304 | 0.289 | 0.334 |
| **CGP (Llama 2–7B)** | **0.315** | **0.300** | **0.347** |
| **CGP (Llama 4 Scout 17B)** | **0.321** | **0.307** | **0.353** |
| **CGP (Llama 3.3 70B)** | **0.328** | **0.313** | **0.359** |

*Exposure-bias Controlled Evaluation (Llama 3.3 70B)*

| Setting | GA-LLM | CGP | $\Delta$ |
|---|---|---|---|
| Near POIs ($\leq$1km) | 0.312 | 0.318 | +0.006 |
| Far POIs ($>$5km) | 0.201 | 0.238 | +0.037 |
| Popular POIs (Top 20%) | 0.342 | 0.349 | +0.007 |
| Long-tail POIs (Bottom 80%) | 0.187 | 0.224 | +0.037 |

(b) Case study (RQ3)

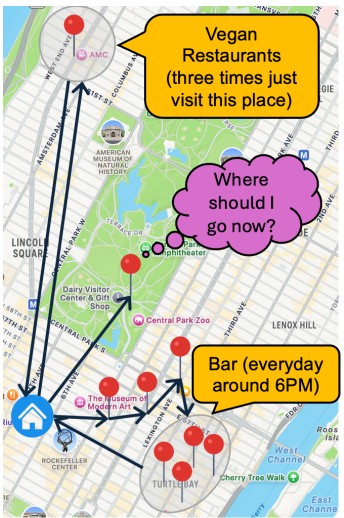

## 5.2 PERFORMANCE COMPARISON (RQ1)

To assess the effectiveness of our proposed CGP, we conduct extensive experiments as shown in Table 2. CGP consistently outperforms all baselines across datasets. With the strongest backbone Llama 3.3 70B, CGP achieves Acc@1 of 0.328 on Gowalla, 0.313 on NYC-T, and 0.359 on Ma-ST, compared with 0.304, 0.289, and 0.334 for GA-LLM. The gains are most evident on Gowalla and NYC-T, where shorter trajectories and skewed POI distributions make baselines more prone to overfitting nearby check-ins. To further assess if these gains are associated with exposure bias, we conducted stratified evaluations averaged across the three datasets. Test cases were divided by distance into nearby ($\leq$1km) and distant ($>$5km) categories, and by popularity into top 20% most-visited POIs and the remaining long-tail 80%. The gap between CGP and GA-LLM remains small for nearby and popular POIs but grows significantly for distant and long-tail POIs, where exposure bias is strongest. This contrast shows that LLMs alone, while improving semantic coverage, cannot overcome the spurious correlations induced by spatial convenience. Overall, the causal adjustments in CGP reduce exposure-induced distortions and enhance the robustness of POI prediction.

## 5.3 ABLATION STUDY (RQ2)

To assess the contribution of each component, we conduct ablation experiments as shown in Table 3b. (1) **w/o GPE**: removes geographic propensity estimation, eliminating causal balancing; (2) **w/o CGS**: removes causal-geographic similarity, using only cosine similarity; (3) **w/o ECR**: removes exposure-consistent retrieval, relying solely on semantic similarity. Table 3b reports the Acc@1 results for CGP with the Llama 3.3 (70B) backbone. The largest accuracy drop occurs when geographic propensity estimation is removed, confirming that causal balancing is the most critical driver of performance improvements. Removing causal-geographic similarity or exposure-consistent retrieval also degrades accuracy, though to a smaller degree, since they ensure retrieval is aligned in both intent and exposure. Together, these findings make clear that LLMs contribute semantic richness, but the causal modules provide the essential bias correction. The superior performance of CGP therefore stems directly from combining the two, rather than scaling LLM capacity alone.

## 5.4 INTERPRETABILITY ANALYSIS (RQ3)

To illustrate how CGP enhances interpretability, we analyze a real case from the NYC-T dataset (Figure 2b). User 882 often checks in at nearby Bars around 6 PM after work in midtown Manhat-

Table 3: (a) Sensitivity analysis (RQ4) and (b) ablation study (RQ2) at Acc@1.

(a) Sensitivity analysis (RQ4)

| Setting | Value | Gowalla | NYC-T | Ma-ST |
|---------|-------|---------|-------|-------|
|         | 1     | 0.316   | 0.300 | 0.347 |
|         | 3     | 0.324   | 0.308 | 0.353 |
| $k$     | 5     | **0.328** | **0.313** | **0.359** |
|         | 7     | 0.327   | 0.311 | 0.358 |
|         | 10    | 0.325   | 0.309 | 0.356 |
|         | 0.0   | 0.320   | 0.305 | 0.351 |
|         | 0.5   | **0.328** | **0.313** | **0.359** |
| $\lambda$ | 1.0 | 0.327   | 0.312 | 0.358 |
|         | 1.5   | 0.324   | 0.310 | 0.356 |
|         | 2.0   | 0.321   | 0.308 | 0.354 |

(b) Ablation study (RQ2)

| Variant | Gowalla | NYC-T | Ma-ST |
|---------|---------|-------|-------|
| CGP (full) | **0.328** | **0.313** | **0.359** |
| w/o GPE | 0.314 | 0.298 | 0.346 |
| w/o CGS | 0.320 | 0.304 | 0.351 |
| w/o ECR | 0.322 | 0.307 | 0.353 |

tan, a routine strongly shaped by spatial convenience. This user also makes repeated trips to distant Vegan Restaurants, which, although less accessible, reflect a stable dietary preference. Conventional models emphasize the frequent bar visits and would predict another bar, thus mistaking convenience-driven exposure for preference. CGP avoids this through two mechanisms. First, geographic propensity scores estimate whether a visit is driven by convenience: bars near home have high values (e.g., $p \approx 0.82$), while vegan restaurants have lower values (e.g., $p \approx 0.37$), highlighting them as stronger preference indicators. Second, causal-geographic similarity ensures that retrieval selects trajectories consistent not only in semantics but also in exposure; under the best setting ($k = 5, \lambda = 0.5$), it reinforces the pattern of traveling farther for vegetarian cuisine observed in similar users. By combining these components, CGP predicts the vegan restaurants as the next destination and, importantly, explains its reasoning: nearby bars are down-weighted as high-propensity, convenience-driven visits, while distant vegan restaurants are emphasized as preference-driven signals.

## 5.5 HYPERPARAMETER SENSITIVITY (RQ4)

We evaluate CGP's robustness with respect to two key parameters: (1) the number of retrieved trajectories $k$ in exposure-consistent retrieval, and (2) the penalty weight $\lambda$ in causal-geographic similarity. As shown in Table 3a, performance improves as $k$ increases from 1 to 5, since additional exposure-consistent examples enrich the behavioral context. Beyond $k = 7$, however, accuracy plateaus and then drops slightly at $k = 10$, suggesting that overly broad retrieval introduces irrelevant or noisy trajectories. This dilutes causal alignment and risks reintroducing spurious correlations, indicating that moderate retrieval depth achieves the best trade-off between context richness and causal precision. For $\lambda$, setting the penalty to zero ($\lambda = 0$) consistently underperforms, confirming that exposure alignment is essential to suppress proximity-driven bias. The best results are achieved at $\lambda = 0.5$, where semantic similarity and exposure consistency are balanced. Larger values ($\lambda \geq 1.5$) reduce accuracy slightly by over-penalizing exposure differences and discarding useful examples. Overall, CGP remains robust across a wide parameter range, with optimal tuning ($k = 5, \lambda = 0.5$) consistently yielding the highest performance.

## 6 CONCLUSION AND FUTURE WORK

In conclusion, we presented CGP, a unified framework that integrates causal inference with LLM-based trajectory modeling to address geographic exposure bias in POI prediction. We designed CGP to disentangle genuine preferences from proximity-driven behaviors. Our model combines geographic propensity estimation with causal-geographic similarity and exposure-consistent retrieval. This method achieved consistent improvements across diverse LBSN datasets.. For our future work we have three main goals. First we will incorporate multi-modal contextual signals. We also plan to design dynamic causal adjustments for evolving mobility patterns. Finally we aim to extend CGP to broader applications like location-based recommendation and urban mobility forecasting.

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
