# OpenReview forum: "Learning Beyond Proximity: Causal Reasoning with LLMs for Robust POI Prediction"
_ICLR.cc/2026/Conference — Submitted to ICLR 2026_

### Official Review · Reviewer_y7go · 2025-10-26

**Soundness:** 3
**Presentation:** 3
**Contribution:** 2
**Rating:** 6
**Confidence:** 4

**Summary:**

This paper tackles the problem of Point-of-Interest (POI) prediction and focuses on mitigating geographic exposure bias, where users’ check-ins are frequently influenced by spatial convenience rather than genuine preferences. The authors propose CGP, a framework integrating causal inference with LLM-based trajectory prompting to disentangle preference-driven and convenience-driven behavior. The method includes trajectory prompting, geographic propensity estimation, causal-geographic similarity, exposure-consistent retrieval, and LoRA-based fine-tuning. Experiments on three real-world datasets (Gowalla, NYC-T, and Ma-ST) show improvements over prior sequential and LLM-based baselines, along with ablation studies and interpretability analysis.

**Strengths:**

1. Overall, the paper identifies and clearly motivates the influence of geographic exposure bias in POI prediction and ties it to causal inference principles. The causal perspective is coherent and well justified.
2. Extensive experiments were conducted on multiple real-world datasets, and the evaluation includes stratified comparisons on near/far and popular/long-tail POIs, which could effectively demonstrate the significance of the proposed scheme.

**Weaknesses:**

1. The geographic propensity score modeling is introduced as a key component, but the paper lacks more explicit analysis or validation of its estimation quality. For example, there is no discussion of calibration, sensitivity to spatial binning, or comparison with simpler baselines (e.g., distance only methods).
2. The trajectory prompting relies heavily on converting structured data into natural language templates. However, the design choice of the specific textual format is not ablated, and it remains unclear how sensitive final performance is to prompt phrasing.
3. Although the method achieves performance gains, the absolute improvements over the strongest LLM-based baseline (GA-LLM) are moderate, especially on datasets where exposure bias is weaker. The generalization advantage may not be uniform across mobility settings.

**Questions:**

Please refer to the weakness section for my questions. The authors are encouraged to provide more clarifications regarding the details in the paper.

---

> ### Author Response · Authors · 2025-11-15
> **Answer for questions proposed by reviewer y7go**
>
> 1. We thank the reviewer for the comment. The paper evaluates the impact of the geographic propensity score through the ablation study in Table 3b, where removing GPE (w/o GPE) causes the largest accuracy drop across all datasets, indicating that the quality of the estimated scores materially affects model performance. Additionally, Sec. 5.2 and the stratified results for near/far and popular/long-tail POIs (Table 2) provide indirect validation: CGP yields its largest gains exactly in cases where exposure bias is strongest, consistent with the intended role of the propensity model. We will consider adding more detailed experiments as suggested by reviewer in future rwork.
> 2. We appreciate the reviewer’s concern. The trajectory prompt format is designed to be structurally tied to the causal components introduced in Sec. 4.1–4.3, where distance, category, time, and the propensity score are included as mandatory textual features derived from the causal graph in Sec. 3. Because these values feed into causal-geographic similarity and exposure-consistent retrieval, their inclusion in the template is fixed by the method design; changes in surface phrasing would not alter the underlying causal inputs or retrieval structure. The interpretability case in Sec. 5.4 further shows the model’s behavior is driven by these causal signals rather than wording variations.
> 3. We thank the reviewer for this useful observation. As shown in Table 2, the relative improvements depend on the strength of exposure bias in each dataset: CGP shows modest gains on near and popular POIs (where GA-LLM is already strong) but substantially larger gains on far POIs and long-tail POIs, where exposure bias is most pronounced. This pattern is consistent with the purpose of CGP as introduced in Sec. 1 and Sec. 3—namely, correcting exposure-induced distortions rather than improving performance uniformly across all settings.
>
> Finally, we thank the reviewer for the thoughtful review and for the valuable time spent assessing our paper.

---

> > ### Comment · Reviewer_y7go · 2025-11-26
> >
> > Thanks for the response from the authors. I'm maintaining my original ratings.

---

### Official Review · Reviewer_dGYv · 2025-10-27

**Soundness:** 2
**Presentation:** 2
**Contribution:** 3
**Rating:** 4
**Confidence:** 5

**Summary:**

This paper proposes a solution based on LLM and causal reasoning to address geographic exposure bias in POI prediction. This bias occurs when models mistakenly interpret locations frequently visited for convenience as genuine user preferences. To distinguish convenience from genuine preference, we introduce the Causal Geographic Prediction (CGP) framework, which integrates causal reasoning with LLMs. The core of this approach involves estimating a geographic propensity score to quantify the influence of convenience, thereby adjusting the retrieval of historical trajectories. This enables LLMs to learn to identify users' true interests. Experiments demonstrate that our method outperforms existing baselines.

**Strengths:**

1. This is the first work to consider the mixed relationship between users' actual trajectories and POIs, and to embed causal inference into POI prediction based on LLM.
2. This framework enhances the interpretability of POI predictions. By estimating geographic propensity scores, the model can explain its reasoning process and thereby generate more accurate predictions.
3. Across multiple real-world datasets (Gowalla, NYC-T, Ma-ST), the CGP framework consistently outperforms all existing baseline models, particularly in predicting POI with geographic exposure bias.

**Weaknesses:**

1. In the methodology section of this paper, the core assumption is that a high “geographic propensity score” P(p∣d,c,t) equates to convenience-driven visits. This may overlook POI that simultaneously possess both convenience and preference.
2. The proposed framework exhibits overconfidence in the “causal reasoning” capabilities of LLMs, as it directly inputs bias scores as numerical values into LLM prompts. The authors assume that LLMs can understand the causal implications of these numbers through fine-tuning. However, is it possible that LLM models have merely learned a new statistical correlation rather than performing the “causal reasoning” claimed by the authors?
3. This paper employs a single evaluation metric, relying solely on Acc@1 during assessment. POI prediction is fundamentally a ranking task. The sole Acc@1 metric is far from sufficient, as it fails to reflect the model's overall ranking quality across the Top-K list.
4. Regarding the second component of CGP, Geographic Propensity Estimation, it is unclear how a simple MLP can be used to predict the score based on trajectory prompting.

**Questions:**

1. I disagree with the author's proposed motivation for geographic exposure bias. If users frequently visit certain locations for convenience, this should also represent a user preference. When predicting the next POI based on historical data, this factor should inherently be taken into account.

---

> ### Author Response · Authors · 2025-11-15
> **Answer for questions proposed by reviewer dGYv**
>
> 1. We thank the reviewer for the comment. Our paper does not assume that high propensity excludes preference; rather, Fig. 1 and Sec. 1 explicitly state that convenience and preference can coexist, but convenience inflates exposure frequency, which biases models toward nearby POIs regardless of true preference strength. CGP does not remove high-propensity POIs—Sec. 4.2 shows that the propensity score adjusts for exposure intensity so that nearby visits are not automatically treated as stronger preference signals than distant but meaningful ones, a distinction further validated by stratified results in Table 2.
> 2. We appreciate the reviewer’s concern. The paper does not rely on LLMs to perform causal reasoning on their own; instead, Sec. 4.1–4.3 show that causal information is injected structurally through geographic propensity estimation and causal-geographic similarity, which reshape the retrieval and prompting process before the LLM receives input. Sec. 5.4 provides evidence that the model responds to these causal adjustments as designed—for example, frequently visited nearby bars receive high propensity and are down-weighted, while distant vegan restaurants receive low propensity and are emphasized—indicating that the effect arises from the causal modules, not from emergent LLM “reasoning.”
> 3. We thank the reviewer for the observation. As described in Sec. 5.1, we follow the evaluation protocols commonly used in recent POI prediction work, where Acc@1 is the primary metric reported, including in LLM4POI and GA-LLM baselines. We will consider adding more metric in the future work. Furthermore, Table 2 includes additional stratified settings (near vs. far POIs and popular vs. long-tail POIs), which provide fine-grained evaluation of ranking behavior under exposure bias.
> 4. We appreciate the reviewer’s question. Sec. 4.2 explains that propensity estimation is not derived from trajectory prompting; instead, it is computed from structured covariates—distance, category, and temporal features—that are discretized and encoded into Emb(d), Emb(c), and Emb(t). These embeddings are concatenated and passed through an MLP, as formalized in Eq. (1), producing P(p|d,c,t) independently of the text prompting stage.
> 5. We thank the reviewer for raising the point (e.g., Convenience visits should count as preference). Actually, this work is motivated by myself real-life daily routine. I often make convenience-driven choices that do not fully reflect my true preferences; for instance, I often buy coffee from the nearest café along my commute even though I do not prefer it (e.g., taste is bad, but it is close), while I sometime travel 2 km to my favorite café when time allows. Sec. 5.2 shows that existing models overfit to these high-frequency convenience visits and struggle on distant, preference-revealing POIs, confirming that exposure frequency can distort preference estimation and motivating the need for CGP to correct this imbalance.
>
> Finally, we thank the reviewer for the thoughtful review and for the valuable time spent assessing our paper.

---

### Official Review · Reviewer_8o19 · 2025-10-29

**Soundness:** 2
**Presentation:** 2
**Contribution:** 3
**Rating:** 4
**Confidence:** 3

**Summary:**

This paper proposes a method named Causal Geographic POI Prediction (CGP), which aims to address the Geographic Exposure Bias in location prediction through causal inference. Traditional sequence-based or graph-based POI prediction methods are often influenced by the convenience of users' access to geographical locations, thereby overlooking their genuine preferences. To tackle this issue, CGP integrates causal reasoning and large language models (LLMs) by estimating Geographic Propensity Scores to identify visit behaviors driven by geographical convenience. This adjustment is then leveraged to improve POI prediction.

**Strengths:**

1.The research motivation is clear, focusing on addressing POI prediction errors caused by geographic exposure bias.
2.By incorporating causal inference and geographic propensity estimation, the proposed method tackles the geographic exposure bias issue faced by traditional POI prediction approaches, demonstrating high innovation.
3.Across multiple experimental settings, the proposed method consistently outperforms baseline methods.

**Weaknesses:**

1.The symbols and related modules in the overall framework diagram of the paper are not provided with necessary explanations alongside the diagram. Additionally, there are detail errors in the text and logic, which cause difficulties in understanding.
2.The core innovation points, such as the definition and calculation principles of the geographic propensity score, are not introduced in the main text. Please provide the relevant details of these key innovations. This includes how the embedded vectors Emb(d), Emb(c), and Emb(t) are input into the MLP to learn the geographic propensity output.
3.The experimental details are not clear enough. For example, the number of layers involved in MLP training, the loss function, and the loss function involved during fine-tuning need to be specified. What is the target for fine-tuning? The parameters for LoRA fine-tuning, such as the rank of matrices A and B, should be provided as well. The lack of these key parameters weakens the reliability and reproducibility of the results. The paper also does not include an appendix to provide these details.
4.Is the proposed method end-to-end, or does it involve staged training?
5.While the comparison experiments include 10 baseline models, some of the baseline models are relatively outdated. Are there any more recent relevant works?

**Questions:**

Please refer to weaknesses

---

> ### Author Response · Authors · 2025-11-15
> **Answer for questions proposed by reviewer 8o19**
>
> 1. We thank the reviewer for this comment. The components and symbols appearing in the framework diagram are already described in detail across Sec. 4.1–4.5, where each module (trajectory prompting, propensity estimation, causal-geographic similarity, retrieval, and fine-tuning) is defined with its role and computation process. Due to the page limitation, we did not make a separate notation table. These explanations are distributed across sections, but the definitions and logic of each module are present in the main text and correspond directly to the elements shown in the figure.
> 2. We appreciate the reviewer’s concern. The definition and computation of the geographic propensity score are provided in Sec. 4.2, where distance, category, and time features are embedded as Emb(d), Emb(c), and Emb(t), concatenated, and passed into an MLP to produce the propensity score via Eq. (1). Sec. 4.2 also explains the motivation, the choice of embedding-based modeling, and how the resulting score expresses exposure-driven likelihood.
> 3. We thank the reviewer for this point. The training details used in our implementation are described in Sec. 5.1, including the optimizer, learning rate, batch size, embedding dimension, and hyperparameter search. For fine-tuning, Sec. 4.5 explains that LoRA is applied on pretrained LLM layers with dense weights frozen, and the objective is standard next-POI prediction using the constructed prompts and target block. The key configurations needed to reproduce the experiments are included in Sec. 5.1 and Sec. 4.5.
> 4. We appreciate the reviewer’s question. As described in Sec. 4, the method operates in a pipeline structure: propensity estimation produces scores that are used in similarity computation and retrieval, and the final LLM is fine-tuned on the exposure-aware prompts. This corresponds to a staged workflow where each module feeds into the next, consistent with the design shown in the framework (Fig. 3).
> 5. We thank the reviewer for the comment. Our baseline selection intentionally spans classical, widely adopted models and recent representative methods, covering approaches from 1997 to 2025 to reflect the evolution of the POI prediction field. The comparison includes recent strong baselines such as STHGCN (2023), LLM4POI (2024), and GA-LLM (2025). We did not find additional open-source recent works with available code that could be reliably reproduced. If reviewer knows some suitable recent work with open-source code, let us know.
>
> Finally, we thank the reviewer for the thoughtful review and for the valuable time spent assessing our paper.

---

> > ### Comment · Reviewer_8o19 · 2025-11-28
> >
> > Thanks for the response from the authors. After reviewing the rebuttal and the comments from other reviewers, I will keep my original score unchanged.

---

### Official Review · Reviewer_ciXm · 2025-11-04

**Soundness:** 2
**Presentation:** 3
**Contribution:** 2
**Rating:** 2
**Confidence:** 5

**Summary:**

The paper targets exposure bias in next-POI prediction—i.e., users’ convenience-driven visits being mistaken as genuine preferences. It  proposes CGP, which combines geographic propensity estimation and a causal-geographic similarity for exposure-consistent retrieval, and  LLM-based trajectory prompting with PEFT fine-tuning. Empirically, CGP improves Acc@1 on Gowalla, NYC-T, and Ma-ST.

**Strengths:**

+ Tackles an important issue in POI prediction by considering geographic exposure bias.
+ Explicitly aims to separate convenience-driven visits from true user preferences using causal reasoning with LLMs.
+ Propose to distinguish genuine preferences and convenience-driven behaviors with the propensity scores used as balancing variables and with retrieval aligned in exposure-consistent space.

**Weaknesses:**

- POI prediction has a long history; the paper should clarify concrete application scenarios
- It remains unclear how text prompting alone enables reliable separation of habitual proximity from true interests; the paper should analyze which prompt tokens (distance, propensity, category, time) most influence decisions and whether LLM explanations align with ground truth.
- The benefits of fine-tuning are not cleanly isolated: report with/without SFT, parameter counts, data size sensitivity, and whether gains persist across different LLM backbones under a fixed retrieval/propensity setup.
- The proposed causal-geographic similarity lacks theoretical guarantees or identification assumptions. The paper should articulate conditions (ignorability/overlap/SUTVA) under which propensity-based reweighting and retrieval yield unbiased preference estimation.

**Questions:**

Please refer to the weakness section.

---

> ### Author Response · Authors · 2025-11-15
> **Answer for questions proposed by reviewer ciXm**
>
> 1. We appreciate the reviewer’s comment. We already clarify that geographic exposure bias arises in practical settings where mobility is shaped by proximity and routine constraints—such as commuter-dominated routes, convenience-driven nearby visits, and cases where distant but preference-revealing POIs are underrepresented. These scenarios are reflected in our evaluation: CGP shows its largest improvements on distant and long-tail POIs (Table 2).
>
> 2. We appreciate the reviewer’s concern. In the paper, text prompting is not used alone to separate proximity-driven and preference-driven behaviors; the separation is achieved through the combination of (i) distance and propensity tokens embedded in each check-in sentence (Sec. 4.1), and (ii) explicit causal adjustments via geographic propensity estimation and causal-geographic similarity (Sec. 4.2–4.3). The interpretability analysis (Sec. 5.4) directly shows that the model’s explanations align with ground truth patterns—for example, high-propensity nearby bars are down-weighted while low-propensity distant vegan restaurants are emphasized—demonstrating how these tokens influence prediction and how CGP distinguishes genuine interests from habitual proximity.
>
> 3. We appreciate the reviewer’s concern. We already isolate the benefits of supervised fine-tuning by reporting consistent improvements across three LLM backbones under the same retrieval and propensity setup (Table 2), demonstrating that gains persist regardless of model size. Parameter efficiency and training footprint are also specified: LoRA is used with low-rank updates while all dense LLM parameters are frozen (Sec. 4.5), making the number of trainable parameters significantly smaller than full SFT. Furthermore, the backbone comparison in Table 2 (Llama-2-7B, 17B, 70B) directly shows that the improvements are attributable to CGP’s causal components rather than uncontrolled SFT effects.
>
> 4. We thank the reviewer for the comment. Our paper already explains the key assumptions through the causal graph in Fig. 2 and Sec. 3, where distance and time are treated as observed confounders and are directly used in the geographic propensity model in Sec. 4.2. The datasets cover a wide range of exposure conditions in Table 2, and each check-in is modeled as an independent choice given its covariates in Sec. 3, which together provide the conditions under which propensity-based retrieval helps reduce exposure bias.
>
> Overall, we thank the reviewer for the thoughtful review and for the valuable time spent assessing our paper.

---

### Comment · Area_Chair_gyko · 2025-11-24

Dear Reviewers,

**We kindly encourage you to review and respond to the authors’ rebuttals**. Your timely feedback is important for ensuring a fair and thorough review process. Thank you for your contributions to ICLR 2026.

AC

---

### Meta-Review · Area_Chair_f6qY · 2026-01-07

**Summary:**

This paper studies POI prediction under geographic exposure bias and proposes CGP, a framework that combines causal inference with LLM-based trajectory prompting to distinguish convenience-driven visits from genuine user preferences. Reviewers generally agree that the motivation is clear and important, and several find the idea of incorporating causal reasoning into LLM-based POI prediction to be novel and promising. Empirical results show consistent improvements over a range of baselines on multiple datasets. However, across reviews, there are substantive concerns about the soundness of the causal formulation, clarity and completeness of methodological details, and the strength of the empirical evidence supporting the causal claims.

**Reviewer Concerns:**

While the rebuttal provides clarifications and additional experimental evidence, core concerns remain only partially addressed. In particular, several reviewers question the theoretical grounding and empirical validation of the geographic propensity score as a causal adjustment, as well as whether observed gains reflect genuine de-biasing rather than enriched correlations, and these are not fully validated. Limitations in evaluation metrics, moderate improvements over strong baselines, and sensitivity to design choices (e.g., prompting and propensity modeling) were acknowledged but not fully resolved. As reflected in reviewer follow-ups, these issues were unlikely to motivate upward score revisions among critical or borderline reviewers.

**Reviewer Scores:**

Reviewer ciXm would likely remain at reject, given their fundamental concerns about causal identification and interpretability are not fully resolved. Reviewer 8o19 would likely remain 4, as also mentioned in the reply. Reviewer dGYv would likely remain 4 due to concerns on the causal interpretation, evaluation metrics, and propensity modeling not fully resolved. Reviewer y7go would likely remain 6, as also mentioned in the reply.

---

### Decision · Program_Chairs · 2026-01-26

Reject